# Relations between Shadowing and Inverse Shadowing in Dynamical Systems

**Alexey A. Petrov**

Faculty of Mathematics and Mechanics, Saint-Petersburg State University, 198504 Saint Petersburg, Russia; al.petrov239@gmail.com

**Abstract:** In this paper, we study relations between shadowing and inverse shadowing for homeomorphisms of a compact space. We present an example of a smooth diffeomorphism of a compact three-dimensional manifold that has the shadowing property and does not have the inverse shadowing property. For some classes of inverse shadowing, we construct examples of homeomorphisms that have the inverse shadowing property but do not have the shadowing property.

**Keywords:** shadowing property; inverse shadowing property; axiom A; structural stability; strong transversality condition; $C^0$-transversality condition

---

## 1. Introduction

In the theory of dynamical systems, a lot of publications are devoted to the problems of shadowing of pseudotrajectories and of inverse shadowing (let us mention the monographs [1,2] and also the papers [3,4]).

At present, we do not have a complete answer to the question: How are these two properties related?

In this paper, we give a review of the existing answers to the above question and extend the list of examples which show that the two properties are nonequivalent.

Let us formulate the main definitions which we need.

Let $f: M \to M$ be a homeomorphism of a metric space $(M, \text{dist})$.

**Definition 1.** *Fix a $d > 0$. We say that a sequence of points $\xi = \{\xi_i \in M \mid i \in \mathbb{Z}\}$ is a $d$-pseudotrajectory of $f$ if the inequalities*

$$\text{dist}(f(\xi_i), \xi_{i+1}) < d$$

*hold for all $i \in \mathbb{Z}$.*

**Definition 2.** *We say that $f$ has the shadowing property if for any $\varepsilon > 0$ there exists a $d = d(\varepsilon) > 0$ such that for any $d$-pseudotrajectory $\xi = \{\xi_i\}$ one can find a point $p \in M$ such that the inequalities*

$$\text{dist}(f^i(p), \xi_i) < \varepsilon$$

*hold for any $i \in \mathbb{Z}$. In this case, we say that the point $p \in M$ $\varepsilon$-shadows the pseudotrajectory $\xi$.*

We also need a stronger variant of Definition 2.

**Definition 3.** *If there exist constants $L, d_0 > 0$ and $\gamma \in (0, 1)$ such that for any $d$-pseudotrajectory $\xi$ of $f$ with $d \in (0, d_0)$ one can find a point $p \in M$ that $Ld^\gamma$-shadows the pseudotrajectory $\xi$, we say that the homeomorphism $f$*

has the Hölder shadowing property with Hölder exponent $\gamma$. If $\gamma = 1$, we say that the homeomorphism $f$ has the Lipschitz shadowing property.

Now, we define the inverse shadowing property for a homeomorphism of a metric space. There exist several various definitions of this property (see, for example, [1,5]).

For definiteness, in this paper, we only consider the class $\Theta_s(f, d)$ of $d$-methods (for details, see [4,5]).

**Definition 4.** *Fix a number $d > 0$. We say that a sequence of continuous mappings*

$$g_n \colon M \to M, \quad n \in \mathbb{Z} \tag{1}$$

*is a continuous $d$-method for $f$ if the inequalities*

$$\mathrm{dist}(f(g_{n-1}(x)), g_n(x)) < d \tag{2}$$

*hold for any $n \in \mathbb{Z}$ and any point $x \in M$.*

**Definition 5.** *We say that a homeomorphism $f$ has the inverse shadowing property if for any $\varepsilon > 0$ there exists a $d > 0$ such that for any continuous $d$-method $\{g_k\}$ and for any point $x \in M$ one can find a point $p \in M$ for which the following inequalities hold:*

$$\mathrm{dist}(g_k(p), f^k(x)) < \varepsilon, \quad k \in \mathbb{Z}. \tag{3}$$

Similarly to Definition 3, one can define Hölder and Lipschitz inverse shadowing properties.

Naturally, there arises the question of distinguishing systems that satisfy Definitions 2 and 5 and their Lipschitz analogs.

Let us start with a survey of the corresponding results.

First, we formulate definitions of two properties that are closely related to the shadowing and inverse shadowing properties: Axiom A and the strong transversality condition.

**Definition 6.** *A diffeomorphism $f \colon M \to M$ of a smooth manifold $M$ satisfies Axiom A if its nonwandering set $\Omega(f)$ is hyperbolic and periodic points are dense in this set:*

$$\mathrm{Clos}(Per(f)) = \Omega(f).$$

It is well known that for any point $p$ of a hyperbolic set $K \subseteq M$ of a diffeomorphism $f$ there exist its stable and unstable manifolds $W^s(p)$ and $W^u(p)$ that are smoothly embedded disks of complementary dimensions,

$$\dim(W^s(p)) + \dim(W^u(p)) = \dim M.$$

The strong transversality condition is formulated as follows.

**Definition 7.** *Let $f \colon M \to M$ be a diffeomorphism of a smooth manifold that satisfies Axiom A. We say that $f$ satisfies the strong transversality condition if for any points $p_1, p_2 \in \Omega(f)$, any point $q \in W^s(p_1) \cap W^u(p_2)$ is a point of transverse intersection of the manifolds $W^s(p_1)$ and $W^u(p_2)$.*

The following theorem has been proved in a series of papers [1,6–8].

**Theorem 1.** *Let $f \colon M \to M$ be a diffeomorphism of a smooth closed manifold $M$. Then, the following three statements are equivalent:*

(1)   *f has the Lipschitz shadowing property;*
(2)   *f has the Lipschitz inverse shadowing property;*
(3)   *f satisfies Axiom A and the strong transversality condition.*

Thus, for a diffeomorphism of a smooth closed manifold, the Lipschitz shadowing property and Lipschitz inverse shadowing property are equivalent.

In the paper [9], it was shown that for diffeomorphisms of two-dimensional surfaces that satisfy Axiom A, shadowing property and inverse shadowing property are equivalent. Namely, the following result has been proved.

**Theorem 2.** *Let $f : M \to M$ be a diffeomorphism of a two-dimensional surface that satisfies Axiom A. Then, the following three statements are equivalent:*

(1)   *f has the shadowing property;*
(2)   *f has the inverse shadowing property;*
(3)   *f satisfies the $C^0$ transversality condition.*

Note that the two-dimensional $C^0$ transversality condition used in the paper [9] has a natural multidimensional generalization (see [10]). At the same time, the multidimensional analog of Theorem 2 does not hold; the paper [11] contains an example of a diffeomorphism of a three-dimensional manifold that satisfies Axiom A, has the Hölder shadowing property, and has two hyperbolic fixed points such that their one-dimensional stable and unstable manifolds have a point of intersection (thus, the $C^0$ transversality condition formulated in the paper [10] is violated).

In the next section, we show that for a diffeomorphism that satisfies Axiom A, the $C^0$ transversality condition is necessary for inverse shadowing. Thus, the example constructed in [11] is an example of a diffeomorphism that has the shadowing property and does not have the inverse shadowing property.

Let us also mention the example of the shift homeomorphism on the space of binary sequences $\Sigma_2 = \{0, 1\}^{\mathbb{Z}}$:

$$\sigma : \Sigma_2 \to \Sigma_2,$$

$$\sigma(x)_i = x_{i+1}.$$

It was shown in the paper [12] that the shift homeomorphism has the shadowing property and does not have the inverse shadowing property. However, the peculiarity of this example is the fact that the space $\Sigma_2$ is not a manifold, while in known examples where this space appears as a hyperbolic set of a diffeomorphism (for example, in the Smale horseshoe), the corresponding diffeomorphism has the inverse shadowing property.

If we modify Definition 5 and require that, for a $d$-method, any mapping $g_n : M \to M$ is a homeomorphism, then it follows from the paper [13] that any pseudo-Anosov system on a two-dimensional surface has the inverse shadowing property with respect to such a class of $d$-methods. In Section 2, we give a simpler example of a homeomorphism of a metric space that has the inverse shadowing property with respect to such a class of $d$-methods.

## 2. Shadowing Property Does Not Imply Inverse Shadowing Property

We start this section with the definition of the notion of $C^0$-transversality (see [10]) which we need in what follows.

Let $(M, \text{dist})$ be a connected, smooth, closed manifold with Riemannian metric dist and let $A$ be a topological space.

We endow the space of all continuous mappings from $A$ to the manifold $M$ (which we denote by $C(A, M)$) with the $C^0$-uniform metric defined as follows: for $f_1, f_2 \in C(A, M)$ we set

$$|f_1, f_2|_{C^0} = \sup_{x \in A} (\text{dist}(f_1(x), f_2(x))).$$

**Definition 8.** *Let $\delta > 0$, let $A, B$ be topological spaces, let $U_A \subseteq A$ and $U_B \subseteq B$ be arbitrary subsets, and let $h_1 \colon A \to M$ and $h_2 \colon B \to M$ be continuous mappings. We say that the intersection $h_1(U_A) \cap h_2(U_B)$ is $\delta$-essential if*

$$\widehat{h}_1(U_A) \cap \widehat{h}_2(U_B) \neq \varnothing$$

*for any continuous mappings*

$$\widehat{h}_1 \colon A \to M$$

*and*

$$\widehat{h}_2 \colon B \to M$$

*such that $|\widehat{h}_1, h_1|_{C^0} \leq \delta$ and $|\widehat{h}_2, h_2|_{C^0} \leq \delta$.*

**Definition 9.** *Let, as above, $A, B$ be topological spaces, let $h_1 \colon A \to M$ and $h_2 \colon B \to M$ be continuous mappings, and let $h_1(a) = h_2(b)$ for points $a \in A$ and $b \in B$. We say that the mappings $h_1$ and $h_2$ are $C^0$-transverse at the pair $(a, b)$ if for any open sets $U(a) \subseteq A$ and $U(b) \subseteq B$ such that $a \in U(a), b \in U(b)$ there exists a number $\delta > 0$ such that the intersection $h_1(U(a)) \cap h_2(U(b))$ is $\delta$-essential.*

Let now $f \colon M \to M$ be a diffeomorphism that satisfies Axiom A, let $p, q \in \Omega(f)$, and let $x \in W^s(p) \cap W^u(q)$. Denote by

$$i_{W^s(p)} \colon D^{n_s} \hookrightarrow M$$

and

$$i_{W^u(q)} \colon D^{n_u} \hookrightarrow M$$

the corresponding embeddings of open disks of dimensions $n_s, n_u$, respectively, so that

$$\text{Im}(i_{W^s(p)}) = W^s(p)$$

and

$$\text{Im}(i_{W^u(q)}) = W^u(q).$$

Let also $x_s \in D^{n_s}$ and $x_u \in D^{n_u}$ be the points corresponding to $x \in M$, i.e., such that

$$i_{W^s(p)}(x_s) = i_{W^u(q)}(x_u) = x.$$

**Definition 10.** *We say that the point $x$ is a point of $C^0$-transverse intersection of the manifolds $W^s(p)$ and $W^u(q)$ if the imbeddings $i_{W^s(p)}$ and $i_{W^u(q)}$ are $C^0$-transverse at the pair $(x_s, x_u)$.*

*Finally, we say that $f$ satisfies the $C^0$-transversality condition if for any points $p, q \in \Omega(f)$, any point $x \in W^s(p) \cap W^u(q)$ is a point of $C^0$-transverse intersection of the manifolds $W^s(p)$ and $W^u(q)$.*

Let us note (see [10]) that a point of transverse intersection of two submanifolds is a point of $C^0$-transverse intersection (while, in general, the converse is not true).

The paper [11] contains an example of a diffeomorphism of a three-dimensional manifold

$$f \colon S^1 \times S^2 \to S^1 \times S^2$$

having the following properties:

(1)  $f$ satisfies Axiom A;
(2)  $\Omega(f) = \{p_1, p_2, p_3, p_4\}$, where $p_1$ is a repelling fixed point, $p_4$ is an attracting fixed point, and $p_2$ and $p_3$ are saddle fixed points such that

$$\dim W^u(p_2) = \dim W^s(p_3) = 1;$$

(3)  $W^u(p_2) \cap W^s(p_3) = O(x, f) \neq \varnothing$ (here $O(x, f)$ denotes the trajectory of a point $x$);
(4)  $f$ has the shadowing property.

Since one-dimensional stable and unstable manifolds of fixed points of $f$ have a point of intersection, $f$ does not satisfy the $C^0$-transversality condition.

In the following theorem, we will show that this violation of the $C^0$-transversality condition implies that the diffeomorphism $f$ does not have the inverse shadowing property.

**Theorem 3.** *Diffeomorphism* $f: S^1 \times S^2 \to S^1 \times S^2$ *from [11] has a shadowing property but does not have an inverse shadowing property.*

**Proof.** Our proof follows the corresponding reasoning of the paper [9] (see Lemma 3.1 there). To get a contradiction, let us assume that $f$ has the inverse shadowing property. Let $M = S^1 \times S^2$. For a fixed number $d > 0$, there exists a diffeomorphism $\phi: M \to M$ such that

$$|\mathrm{id}_M, \phi|_{C^1} < d,$$

$$\mathrm{supp}\, \phi \subseteq B(d, x),$$

and

$$\phi(W^s(p_3) \cap B(d, x)) \cap W^u(p_2) = \varnothing.$$

Here, $B(d, x)$ is the open ball of radius $d > 0$ centered at a point $x$.

Set $g = \phi \circ f$. It is easily seen that $g$ is a diffeomorphism that satisfies Axiom A, $\Omega(g) = \{p_1, p_2, p_3, p_4\}$, and

$$W^u_g(p_2) \cap W^s_g(p_3) = \varnothing \tag{4}$$

(here $W^u_g$ and $W^s_g$ are the unstable and stable manifolds of the corresponding fixed points for the diffeomorphism $g$).

Define a $d$-method $\{g_k: M \to M\}$ by the formula

$$g_k = g^k, \quad k \in \mathbb{Z}.$$

Let $\alpha > 0$ be so small that the following statements hold:

(1)  for any point $p \in M$, the inequalities

$$\mathrm{dist}(g^k(p), p_3) < \alpha, \quad k > 0,$$

imply that $p \in W^s_g(p_3)$;
(2)  for any point $p \in M$, the inequalities

$$\mathrm{dist}(g^k(p), p_2) < \alpha, \quad k < 0,$$

imply that $p \in W_g^u(p_2)$.

We get a contradiction; it is easily seen that the trajectory $O(x, f)$ cannot be $\alpha/2$-shadowed by the constructed $d$-method $g_k$ for any $d > 0$.

Indeed, otherwise there exists a point $p \in M$ such that

$$\text{dist}(g_k(p), f^k(x)) < \alpha/2, \quad k \in \mathbb{Z};$$

in this case, the choice of the number $\alpha$ would imply the inclusion $p \in W_g^u \cap W_g^s$ contradicting relation (4).

□

### 3. Inverse Shadowing Property Does Not Imply Shadowing Property

To the best of our knowledge, at present, it is not proved that in the case of a diffeomorphism of a closed manifold, inverse shadowing property does not imply shadowing property.

We devote this section to some partial results in this direction.

**Example 1.** *Our first example in this section is an example of a homeomorphism of a compact metric space that does not have the shadowing property but has the inverse shadowing property in the case where mappings (1) of a d-method are homeomorphisms.*

Thus, let $f \colon S^1 \to S^1$ be a North Pole–South Pole mapping and let $n, s$ be the North Pole and South Pole, respectively (i.e., they are fixed points of $f$, $n$ is a repeller, and $s$ is an attractor). It is well known that $f$ has both the shadowing property and the inverse shadowing property.

Set $M = S^1 \times \{0\} \bigsqcup S^1 \times \{1\} / (s, 0) \sim (n, 1)$. Consider the mapping

$$F \colon M \to M$$

determined by the rule: $F(x, 0) = f(x)$ and $F(x, 1) = f(x)$. Obviously, $F$ does not have the shadowing property.

Clearly, for any homeomorphism $G \colon M \to M$ that is $C^0$-close enough to $F$, the following relations hold:

$$G(S^1 \times \{0\}) = S^1 \times \{0\}$$

and

$$G(S^1 \times \{1\}) = S^1 \times \{1\}.$$

Since $f$ has the inverse shadowing property and any trajectory of $F$ belongs either to $S^1 \times \{0\}$ or to $S^1 \times \{1\}$, the homeomorphism $f$ itself has the inverse shadowing property.

**Example 2.** *Now, we construct an example of a diffeomorphism of a noncompact manifold that does not have the shadowing property but has the inverse shadowing property.*

Consider the Banach space

$$X = \mathbb{R} \times \mathbb{R} \times \mathbb{R}^{\mathbb{Z}}$$

in which the norm is defined as follows: for $v \in X$, where $v = (x, y, \{w_n\}_{n \in \mathbb{Z}})$,

$$|v| = \max \left( |x|, |y|, \sup_{n \in \mathbb{Z}} |w_n| \right).$$

It is convenient for us to denote coordinates of a point $q \in X$ with respect to the representation of $X$ in the form of a direct product:

$$q = (q_x, q_y, q_w),$$

where $q_x, q_y \in \mathbb{R}, q_w \in \mathbb{R}^{\mathbb{Z}}$.

We take as the manifold $M$ on which we define a diffeomorphism $f$ the subset of $X$ which consists of a countable union of disjoint lines $s_m \subset X$, $u_m \subset X$, $m \in \mathbb{Z}$.

We define $f$ so that $f$ maps a line $s_m$ to the line $s_{m+1}$ and a line $u_m$ to the line $u_{m+1}$; $f$ contracts on the lines $s_m$ and expands on the lines $u_m$.

Let $h \colon \mathbb{R} \to [0, 1]$ be a continuous function such that

$$h|_{[-1,1]} \equiv 0$$

and

$$h|_{(-\infty, -2] \cup [2, \infty)} \equiv 1.$$

Introduce the following notation: for $m \in \mathbb{Z}$ set

$$\alpha_m = \sum_{0}^{m} \frac{\text{sign(m)}}{|m| + 1} + \frac{1}{3m + 1}$$

and

$$\beta_m = \sum_{0}^{m} \frac{\text{sign(m)}}{|m| + 1} + \frac{1}{3m + 2}.$$

Define sets $u_m, s_m, m \in \mathbb{Z}$, as follows:

$$s_m = \{(t, \alpha_m, e_{2m}h(t)) \mid t \in \mathbb{R}\}$$

and

$$u_m = \{(t, \beta_m, e_{2m+1}h(t)) \mid t \in \mathbb{R}\}.$$

Here, $e_j$ is the $j$th unit coordinate vector in $\mathbb{R}^{\mathbb{Z}}$.

Set

$$M = \sqcup_{m \in \mathbb{Z}}(u_m \cup s_m).$$

Clearly, $M$ is a complete metric space.

Define $f$ by the formula

- for $x = (t, \alpha_m, e_{2m}h(t)) \in s_m$,

$$f(x) = \left( t/2, \alpha_{m+1}, e_{2(m+1)}h(t/2) \right);$$

- for $x = (t, \beta_m, e_{2m+1}h(t)) \in u_m$,

$$f(x) = \left( 2t, \beta_{m+1}, e_{2(m+1)+1}h(2t) \right).$$

Clearly, the constructed mapping $f$ is a diffeomorphism and the following relations hold for any $n \in \mathbb{Z}$:

$$f(s_n) = s_{n+1}, f(u_n) = u_{n+1}. \tag{5}$$

Let us show that $f$ does not have the shadowing property. For an arbitrary $d > 0$, we construct a $d$-pseudotrajectory that cannot be $1/2$-shadowed by an exact one.

Fix a $d > 0$. Clearly, there exists an $N \in \mathbb{N}$ such that

$$100 \cdot \left(\frac{1}{2}\right)^N < 1$$

and

$$\left| \frac{1}{3N+1} - \frac{1}{3N+2} \right| < d.$$

Define a $d$-pseudotrajectory $\xi = \{\xi_k\}$ as follows. Set

$$\xi_0 = (100, \alpha_0, e_0) \in s_0,$$

$$\xi_i = f^i(\xi_0), \quad i < 0,$$

$$\xi_i = f^i(\xi_0), \quad i = 1, 2, \ldots, N-1,$$

$$\xi_N = f(\xi_{N-1}) + \left(0, \frac{1}{3N+2} - \frac{1}{3N+1}, 0\right) \in u_N,$$

and

$$\xi_{N+1+i} = f^i(\xi_{N+1}), \quad i \in \mathbb{N}.$$

Note that, since $h(t) = 0$ for $|t| \leq 1$, the inclusion $\xi_N \in u_N$ holds. Thus, the sequence $\{\xi_n\}$ is well defined and is a $d$-pseudotrajectory.

To get a contradiction, assume that there exists a point $p = (p_x, p_y, p_w) \in M$ that $1/2$-shadows $\xi$. Then, it is easily seen that the inequality

$$|p - \xi_0| < 1/2$$

implies that $p \in s_0$, and, taking (5) into account, we get the inclusions

$$f^n(p) \in s_n, \quad n \in \mathbb{N}.$$

The last relation for the points

$$f^n(p) = (x_n, y_n, w_n)$$

implies the estimates

$$|x_n| \leq \left(\frac{1}{2}\right)^n \cdot |p_x|$$

for $n \in \mathbb{N}$.

However, since $\xi_N \in u_N$, we also have the relation

$$\lim_{i \to \infty} |(\xi_i)_x| = +\infty.$$

Hence, the pseudotrajectory $\xi$ cannot be $1/2$-shadowed by an exact one.

Let us now show that $f$ has the Lipschitz inverse shadowing property. For that, we show that if $d > 0$ is small enough (namely, if $d < 1/2$), then any $d$-method $\{g_n\}$ satisfies the following relations for all $n, m \in \mathbb{Z}$:

$$g_n(s_m) = s_{n+m}, g_n(u_m) = u_{n+m}. \tag{6}$$

After that, the Lipschitz inverse shadowing property for $f$ follows from (6) and the known fact that a hyperbolic linear mapping on the line $\mathbb{R}$ has the Lipschitz inverse shadowing property.

Relations (6) are easily obtained by induction. For example, let us prove (6) for the sets $s_m$ and for $n \geq 0$:

$$g_n(s_m) = s_{n+m}. \tag{7}$$

Indeed, for $n = 0$ the above equality is obvious. Assume that (7) holds for $n \in \mathbb{N} \cup \{0\}$. Let us show that in this case, (7) holds for $n + 1$. Since the mapping $g_{n+1}$ is continuous and the sets $s_m$ are arcwise connected, the image $g_{n+1}(s_m)$ belongs to one of the sets $u_k$ or $s_k$ for some $k \in \mathbb{Z}$. Let us show that $k = n + m + 1$. Take a point $v \in s_{m+n}$,

$$v = (100, \alpha_{n+m}, e_{2(m+n)}),$$

and an arbitrary point

$$\tilde{v} \in g_n^{-1}(v).$$

Then, $\tilde{v} \in s_m$ by the induction assumption. On the other hand,

$$|g_{n+1}(\tilde{v}) - f(g_n(\tilde{v}))| = |g_{n+1}(\tilde{v}) - f(v)| < 1/2.$$

Since $f(v) = (50, \alpha_{n+m+1}, e_{2(m+n+1)})$, the last relation implies that for the point $g_{n+1}(\tilde{v}) = (\tilde{x}, \tilde{y}, \tilde{w})$,

$$\tilde{w} = e_{2(m+n+1)}.$$

Thus,

$$g_{n+1}(s_m) \subseteq s_{m+n+1}. \tag{8}$$

To prove the converse inclusion, let us consider an arbitrary point

$$p = (x, \beta_{m+n+1}, w) \in s_{m+n+1}.$$

Take two points $p_l, p_r \in s_{m+n+1}$ that lie far to the left and far to the right of $p$, respectively. Namely, set

$$p_l = (x - |x| - 100, \beta_{m+n+1}, e_{2(m+n+1)}),$$

$$p_r = (x + |x| + 100, \beta_{m+n+1}, e_{2(m+n+1)}).$$

Now, take arbitrary points $v_l \in g_n^{-1}(f^{-1}(p_l))$ and $v_r \in g_n^{-1}(f^{-1}(p_r))$ (these points satisfy the relations $f(g_n(v_l)) = p_l$, $f(g_n(v_r)) = p_r$).

We can estimate

$$|g_{n+1}(v_l) - p_l| = |g_{n+1}(v_l) - f(g_n(v_l))| < 1/2, \tag{9}$$

$$|g_{n+1}(v_r) - p_r| = |g_{n+1}(v_r) - f(g_n(v_r))| < 1/2.$$

From (8), it follows that $g_{n+1}(v_l) \in s_{m+n+1}$, and from inequality (9) we conclude that the point $g_{n+1}(v_l)$ also (as well as $p_l$) lies far to the left of $p$.

Similary, $g_{n+1}(v_r) \in s_{m+n+1}$ lies far to the right of $p$.

From the continuity of $g_{n+1}$ it follows that the image $g_{n+1}([v_l, v_r]) \subseteq s_{m+n+1}$ must contain all intermediate values between $g_{n+1}(v_l)$ and $g_{n+1}(v_r)$ (here by $[v_l, v_r]$, we denote the set $\{(x, y, w) \in s_n \mid x \geq (v_l)_x, x \leq (v_r)_x\}$).

Thus, $g_{n+1}([v_l, v_r])$ contains the point $p$. This proves the converse inclusion and completes the induction step.

Thus, we have shown that $f$ has the Lipschitz inverse shadowing property.

**Funding:** This research was funded by the RFBR grant number 18-01-00230a.

**Conflicts of Interest:** The author declares no conflict of interest.

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
