# Peer review of "Relations between Shadowing and Inverse Shadowing in Dynamical Systems"

_axioms, doi:10.3390/axioms8010011_

Round 1
Reviewer 1 Report
The paper deals with homeomorphisms defined on compact metric spaces and study the relations between shadowing and inverse shadowing in such systems. The authors present an interesting example of a diffeomorphism that has the shadowing property, but do not have the inverse shadowing property. They also consider homeomorphisms with some classes of inverse shadowing. The results of the paper are new and interesting. To the best of my understanding, the arguments are correct. I recommend that this article can be published.
Some suggestions:
(1) Page 1, line 11: Replace \[3]; [4]; :::)" by \[3]; [4])";
(2) Page 2, line 40: Cite at least one of the papers;
(3) Page 3, line 78: Why the name of the metric is 1:7em?;
(4) Page 5, line 98: Replace \there). Thus, to get" by \there). To get";
(5) Page 8, line 141: \The converse inclusion is geometrically obvious". It should be written.

Author Response
Response to Reviewer 1 Comments
Point 1: Page 1, line 11: Replace \[3]; [4]; :::)" by \[3]; [4])"
Response 1: fixed.
Point 2: Page 2, line 40: Cite at least one of the papers
Response 2: References to the papers are added.
Point 3: Page 3, line 78: Why the name of the metric is 1:7em?
Response 3: There was a bug with some macros of the journal. Fixed.
Point 4: Page 5, line 98: Replace \there). Thus, to get" by \there). To get"
Response 4: fixed.
Point 5: Page 8, line 141: \The converse inclusion is geometrically obvious". It should be written.
Response 5: Proof of the converse inclusion is added.
Reviewer 2 Report
The shadowing property and the inverse shadowing properte are two different concepts. In the paper, the author cosider that the shadowing property and inverse shadowing property. Actually, he introduce the examples for relationship with the two different notions. I think the notions are very valuable and reasonable to study dynamical systems. So, I recommend to published in the Journal.
Author Response
Dear Reviewer 2,
Thank you for your feedback.
With best regards,
Alexey.
Reviewer 3 Report
see the review report.

Author Response
Response to Reviewer 3 Comments
Point 1: Page 2, line 40. It says that the following theorem has been proved in a series of paper. Please provide a proper reference for the reader
Response 1: References to the papers are added.
Point 2: Page 3, line 78, there is a misprint “1.7em” . It should be d? This also appears several times in other places.
Response 2: fixed (there was a bug with \newcommand and some macros of the journal; "1.7em" should be "dist").
Point 3: It is better to state a Theorem or Proposition in Section 2 for the result ”There is a smooth diffeomorphism of a compact manifold which has the shadowing property but not the inverse shadowing property.”
Response 3: Theorem environment is added (page 5, line 99)
Point 4: Page lines 139 and 140, ”Lipschitz shadowing” should be ”Lipschitz inverse shadowing”?
Response 4: It is true, fixed.